



# A global social activation model of enabling conditions for positive social tipping - the role of sea-level rise anticipation and climate change concern

E. Keith Smith[1*], Marc Wiedermann[2,3,4*], Jonathan F. Donges[5,7], Jobst Heitzig[2], and Ricarda Winkelmann[5,6]

[1]ETH Zürich – International Political Economy and Environmental Politics, Zürich, Switzerland
[2]FutureLab on Game Theory and Networks of Interacting Agents, Complexity Science, Potsdam Institute for Climate Impact Research, Member of the Leibniz Association, Potsdam, Germany
[3]Robert Koch-Institut, Berlin, Germany
[4]Institute for Theoretical Biology, Humboldt University, Berlin, Germany
[5]FutureLab on Earth Resilience in the Anthropocene, Earth System Analysis, Potsdam Institute for Climate Impact Research, Member of the Leibniz Association, Potsdam, Germany
[6]Institute of Physics and Astronomy, University of Potsdam, Potsdam, Germany
[7]Stockholm Resilience Centre, Stockholm University, Stockholm, Sweden
[*]These authors contributed equally to this work.

**Correspondence:** E. Keith Smith (keith.smith@gess.ethz.ch), Ricarda Winkelmann (ricarda.winkelmann@pik-potsdam.de)

**Abstract.** Effective climate change mitigation necessitates swift societal transformations. Social tipping processes, where small triggers initiate qualitative systemic shifts, are potential key mechanisms instigating societal change. A necessary foundation for societal tipping processes is the creation of enabling conditions. Here we assess future sea-level rise estimates and social survey data within the framework of a social activation model to exemplify the enabling conditions for tipping processes. We find that in many countries, climate change concern is sufficient, the enabling conditions and opportunities for social activation already exist. Further, drawing upon the interrelation between climate change concern and anticipation of future sea level rise, we report three qualitative classes of tipping potential that are regionally clustered, with greatest potential for tipping in Western Pacific rim and East Asian countries. These findings propose a transformative pathway where climate change concern increases the social tipping potential, while extended anticipation time horizons can trigger the system towards an alternative trajectory of larger social activation for climate change mitigation.

## 1 Introduction

With increasing greenhouse gas emissions and resultant global warming, the Earth's climate system is becoming more vulnerable to irreversible and abrupt changes (Steffen et al., 2018; Armstrong McKay et al., 2022). The urgency of projected climate impacts is accentuated by interacting tipping elements in the Earth system that, once initiated, carry the potential for cascading 'domino effects' (Lenton and Williams, 2013; Rocha et al., 2018; Wunderling et al., 2021). Rapid societal transformations are



necessary to reduce greenhouse gas emissions and stabilize the Earth's climate system (Otto et al., 2020a; Winkelmann et al., 2022).

Sea-level rise (SLR) presents one of the greatest potential impacts of climate change. With approximately 40% of the world's population residing within 100km of the coastline, SLR poses a global threat to coastal cities, infrastructure, and cultural heritage sites (Nicholls and Cazenave, 2010; Hinkel et al., 2014; Cazenave and Cozannet, 2014; Marzeion and Levermann, 2014). The impacts of SLR are already manifesting today (e.g. increased storm surges, flooding, groundwater salination, and harm to marine ecosystems), and the expected future impacts vary by region (Nicholls et al., 2021; Barragán and de Andrés, 2015). For example, densely populated urban centers in Japan, India, and China are among those potentially most affected by future SLR (Barragán and de Andrés, 2015) (Fig. 1b,c). Depending on the RCP emission scenario, future SLR estimates range between 1m and 5m (Nauels et al., 2017) by the year 2300 (Fig. 1a), where up to 15m SLR by 2300 cannot be ruled out under high emission scenarios (Masson-Delmotte et al., 2021). SLR impacts are further projected to amplify due to the large inertia and positive feedbacks (Garbe et al., 2020), locking in long-term commitments to potentially irreversible SLR (Levermann et al., 2013).

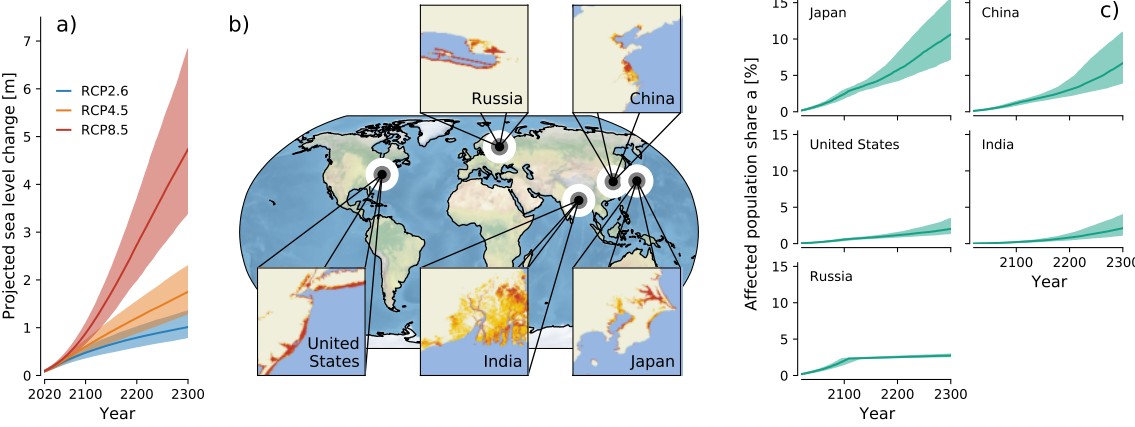

**Figure 1. Projected global mean sea-level rise and affected world regions. (a)** Projected SLR in response to greenhouse gas emissions under different RCP scenarios with low (blue), medium (orange) and high emissions (red) (estimates by MAGICC v2.0). Thick lines indicate the median projected SLR, shaded areas indicate the range that includes 66% of all ensemble runs. **(b)** Regions affected by future SLR in the five countries with the largest share of global greenhouse gas emissions. Lines indicate the areas that are directly affected by future global mean SLR within one (red), four (orange) to seven generations (yellow). **(c)** Average shares of population of the largest national emitters China, United States of America, India, Russia and Japan that is estimated to be directly affected by projected SLR. Shaded areas again indicate the 66% range.

The problem of future sea-level rise presents a unique social-ecological dilemma, a severe clash of time scales. The most serious potential direct impacts of SLR likely manifest on the order of centuries and more (Garbe et al., 2020; Levermann et al., 2013), but mitigation of these impacts necessitates rapid countering societal actions *within the next few years to decades* (Clark et al., 2016). However, future problems are often discounted (Frederick et al., 2002), and human forethought for the future





becomes limited beyond the order of decades (Tonn et al., 2006; Winkelmann et al., 2022). People who do not consider distant future consequences are more likely to have climate skeptical views (Većkalov et al., 2021), and are less likely to engage in mitigating behaviors (Milfont et al., 2012; Corral-Verdugo et al., 2017). Indeed, the dilemma of climate change presents not only a classic tragedy of the commons, but also a *tragedy of the horizon* (Hurlstone et al., 2020).

## 1.1 Social tipping as transformative mechanism for climate actions

Social tipping has been proposed as a mechanism for bringing about necessary sustainability transformations and mitigate climate impacts (Milkoreit et al., 2018; Moser and Dilling, 2007; Otto et al., 2020a; Farmer et al., 2019; Tàbara et al., 2018; Nyborg et al., 2016; Lenton, 2020; Winkelmann et al., 2022). Social tipping is exemplified by qualitative systemic changes resultant from comparatively small changes within the social system, or the broader environment in which the system is embedded (Winkelmann et al., 2022). Similar to the notion of climate tipping processes (Lenton et al., 2008; Armstrong McKay et al., 2022), social tipping dynamics are internally self-amplifying via positive feedback mechanisms such that rapid movements from one qualitative state into another become possible, resulting in sudden large-scale structural changes (Otto et al., 2020a; Milkoreit et al., 2018; Van Ginkel et al., 2020). A necessary element for social tipping processes is the development of 'enabling conditions' Tàbara et al. (2018); Lenton et al. (2022), fostering the likelihood that an event (intentional or not) or a systemic pertubation can instigate rapid, non-linear transformations. The ultimate trigger of social tipping processes can come from forces initiating in both natural or social systems (Winkelmann et al., 2022), as well as those emerging from individual-level changes (Otto et al., 2020a; Milkoreit et al., 2018; Lenton et al., 2022).

Within social-ecological systems, experienced climate impacts (Demski et al., 2017; Konisky et al., 2016), e.g. floods and heat waves (Ricke and Caldeira, 2014), have the potential to shift attitudes and behaviors toward climate change and instigate social tipping processes (Müller et al., 2021). Behavioral changes are more likely if extreme weather events elicit an emotional response, increase the salience of climate change, or when people directly attribute the event to climate change (Sisco, 2021).

Within social systems, extensive empirical studies have focused on identifying drivers of engagement in climate actions (van Valkengoed and Steg, 2019; Bergquist et al., 2022; Cologna and Siegrist, 2020). Specifically, heightened climate change concerns provide a necessary foundation for individual-level engagement in climate actions (van Valkengoed and Steg, 2019; Hoffmann et al., 2022), even though not necessarily serving as an ultimate trigger (Kollmuss and Agyeman, 2002) as concerns are often moderated by other characteristics (such as increased costs (Diekmann and Preisendörfer, 2003), perceptions of individual efficacy (Mayer and Smith, 2019), and trust (Cologna and Siegrist, 2020; Smith and Mayer, 2018)). Notably, the likelihood to engage in mitigating action varies in relation to the perceived psychological distance of climate risks (Spence et al., 2012) such that those perceived to be more culturally, spatially, or temporally proximate increase the likelihood of climate action (Frederick et al., 2002; Brügger et al., 2015; Singh et al., 2017). Accordingly, climate change concern (or risk perceptions) can be best understood as a necessary condition, enabling the likelihood to engage in climate actions - yet on it's own, may not be sufficient to shape individual behavioral change or policy support.



## 1.2 Anticipation of SLR can activate climate actions

In the case of SLR, immediate climate actions for mitigation are required, as even 5-year delays in reaching peak GHG-emissions can each affect a future commitment of $\sim$+0.2m in SLR by 2300 (Mengel et al., 2018). Yet, the most severe immediate impacts of SLR are unlikely to be experienced within current lifetimes, likely between +0.3-1.0m by 2100 (Oppenheimer et al., 2019). Accordingly, *direct experiences* of SLR impacts are unlikely to trigger necessary societal changes on time scales required for mitigation (Pahl et al., 2014), within the remaining 'intervention time horizon' (the period within societal actions can influence whether a natural system tips) (Winkelmann et al., 2022).

Rather, the *anticipation* of the impacts of SLR presents a crucial potential pathway towards bridging the inter-generational gap. Increases to SLR are among the most prominent frames from which people learn about and understand climate impacts (Hamilton et al., 2012). Concern about future SLR can activate desires to leave a positive legacy for the descendants of one's homeland, facilitating engagement in climate actions (Hurlstone et al., 2020). Recent empirical evidence suggests that concerns about future SLR constitutes a unique form of societal climate risk perceptions, increasing support for climate change policies and willingness to engage in pro-climate behaviors by $\sim 15 - 30\%$ (Smith et al., 2020, 2022). Simulation modeling approaches have further noted the importance of anticipation of future climate impacts, where extended time horizons can foster social tipping dynamics towards stabilizing the environment, and potentially preventing social-ecological collapse (Müller et al., 2021). In this case, anticipation SLR can act as an instigating trigger of climate action. When the enabling conditions of a system are such that social tipping processes are increasingly likely, anticipation of climate impacts may be sufficient trigger, bridging the temporal gap between risk exposure and impacts.

Here, we explore this interrelated role of anticipation of SLR and concern for climate change in triggering social tipping processes towards climate action via adoption of a complex contagion social activation model of social tipping. We combine projections of future SLR and cross-national social survey data of climate change concern to estimate likelihoods of social tipping processes (Ashwin et al., 2012), resulting from varying anticipation time horizons and levels of concern in 66 countries. We thereby investigate mechanisms which close the temporal gap between causes and effects of climate impacts, exploring the potential for societal transformations toward climate action and planetary stewardship.

## 1.3 Complex contagion modeling of social tipping

Models of complex social behavioral contagions (Lehmann and Ahn, 2018; Jusup et al., 2021) have explored thresholds for individual action, whereby a Pareto effect of $\sim 25\%$ of the population becoming activated can result in social tipping-like processes of cascading behaviors (Centola et al., 2018). In such cases, changes in individual preference factors (Nyborg et al., 2016) and network structures (Centola and Baronchelli, 2015) can trigger rapid shifts in social norms and behaviors. Complex contagion is commonly simulated using thresholds models (Dodds and Watts, 2004a), social learning and diffusion models, adaptive network models (Gross and Sayama, 2009; Chu et al., 2021), or agent-based voter models (De Marchi and Page, 2014).



Recent advancements have further modeled social tipping dynamics (Müller et al., 2021) – notably, behavioral economic experimental approaches have investigated the difficulty in overcoming perceived costs associated with adopting new norms (Andreoni et al., 2021), agent-based models have identified conditions for rapid adoption of environmental behaviors (Kaaronen and Strelkovskii, 2020), and coupled social-climate models have explored emissions reduction pathways (Moore et al., 2022). Yet, modeling social tipping dynamics, and environmental behavioral change more broadly, remains challenging, as the drivers of human behavior and preference formation are non-deterministic, and are rather the product of an interrelated web of factors (e.g. risk perceptions, costs, social norms, perceived efficacy, trust, political and cultural tastes). Such modeling endeavors are further complicated within cross-cultural settings, as the drivers of climate attitudes and behaviors can vary greatly even between cultural and geographically similar locales (Smith and Mayer, 2019; Marquart-Pyatt et al., 2019).

Given these uncertainties and hetereogenities, we adopt a low-dimensional approach to modeling the interrelated role of concern and anticipation of SLR impacts that is theoretically-based and driven by empirically-derived parameters. We extend a recently developed refinement of Granovetter's threshold model (Granovetter, 1978) for social tipping processes (Wiedermann et al., 2020) that explores engagement in climate actions from cascading contagious dynamics on social networks (Watts, 2002). Climate actions are those taken with the intentionality of mitigating- or adapting to- anticipated, perceived or experienced climate impacts (Stern, 2000). These can encompass a broad range of individual or social behaviors, such as changing consumption patterns, participating in environmental collective actions, and supporting climate change-focused policies and political actors (Tobler et al., 2012).

This approach divides populations into three distinct groups (McCarthy and Zald, 1977; Jenkins, 1983): (I) *certainly active* instigators of climate actions (e.g. opinion shifts, social movements, or adoption of new technologies) , (II) *contingently active individuals* whose characteristics (e.g. norms, beliefs, social identity) broadly align with those of the certainly active population, but who have yet to join these climate actions, and (III) *certainly inactive* individuals who are unlikely to ever join climate actions (e.g. those with norms, values or identities in opposition, or those who lack the capacity to change). Specifically, we adopt a one-dimensional macroscopic approximation of an emergent threshold function which incorporates microscopic network dynamics accounting for the interrelation of behavioral contagion and network structures (see Sect. 3 for details).

## 2 Model design

We apply the complex contagion social activation model of social tipping to the case of SLR, examining how national-level projections of SLR impacts at varying temporal scales (2100, 2200, 2300) and climate change concern affect the share of certainly and contingently active populations across different states world-wide (Fig. 2). For this purpose we use multiple data sources to estimate the necessary parameters and input quantities: (i) future sea-level projections from the MAGICC climate model (Nauels et al., 2017) (Fig. 1a), (ii) global high-resolution topographic (Farr et al., 2007) and population distribution (Center for International Earth Science Information Network, 2016) data, and (iii) pooled, harmonized social survey data on climate change concern from 66 different countries.



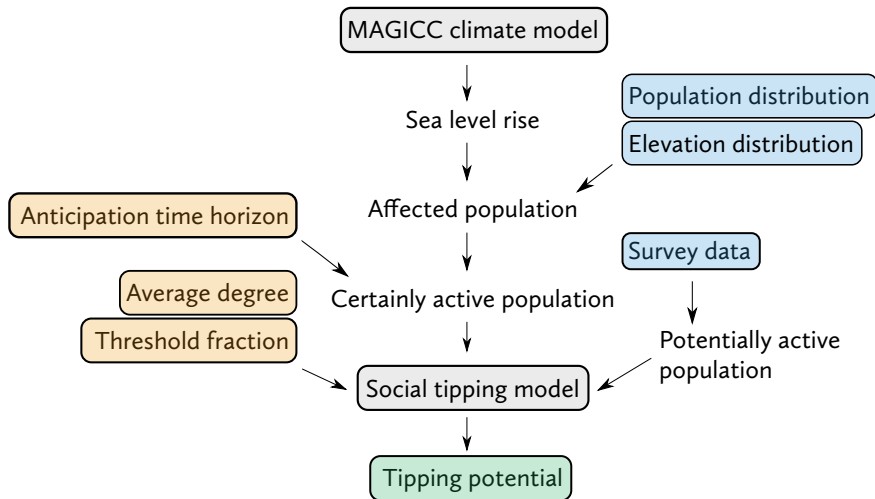

**Figure 2. Visualization of modeling approach.** The tipping potential in a country (green box) is estimated from sea-level projections (based on climate model emulator MAGICC) and the extended Granovetter model of social tipping dynamics. Blue shading indicates external data sources (see Sect. 3). Yellow shading indicates parameters to the model. Core components of the model are marked with grey boxes.

We consider people who live in an area that is likely directly impacted by SLR within a certain *anticipation time horizon*
as certainly active (Group I, Fig. 1c). A country's *potentially active population* (the sum of Groups I and II) with share $p$
is estimated from cross-national survey data on climate change concern (see Tables S1–S3), assuming that higher levels of
concern correspond to larger population shares that can potentially be mobilized for collective climate action. Based on both
group sizes, our model simulates the population share that ultimately participates in an action (see Fig. 3).

The more parsimonious design incorporates a similar level of qualified complexity across all model components and analyt-
ical interpretations, with the goal of minimizing sources of error originating from increased assumptions regarding additional
parameters which remain unknown, or are potentially even unknowable. Similar approaches have further adopted survey data
and low-dimensional modeling designs to explore cross-national social tipping dynamics, particularly in the case of groundwa-
ter management (Castilla-Rho et al., 2017). Accordingly, the modeling goal is to provide grounded, interpretable, qualitative
assessments of the cross-national role of concern and anticipation of SLR, providing a foundation for further exploratory
research in the emerging research on social tipping dynamics.

For every simulation, we expect to find the population share that ultimately engages in climate action in either of three
regimes: (i) an uncritical regime where only the untipped state exists and a large share of the population remains passive, (ii) a
*critical* bi-stable regime where *event-induced tipping* (Ashwin et al., 2012) can move the system into an alternative state (i.e.
potentially active population can be either active or passive, depending on initial conditions), and (iii) the tipped regime that
is reached via *bifurcation-induced tipping* (Ashwin et al., 2012), i.e. where a large share of the potentially active population





is active (see Fig. 3). By using a Monte-Carlo approach, we compute the likelihoods to enter either of the three regimes and denote those as the respective per-country *tipping potentials*. To ensure the robustness of our results, all respective quantities are computed as averages over an ensemble of simulations for random choices of unknown parameters that govern the specific structure of the social tipping model (see Fig. 2 and Sect. 3).

## 3 Detailed model description

### 3.1 Complex contagion model of social tipping dynamics

We adopt a recently developed framework for explaining processes and mechanisms behind social tipping through a network-based microfoundation of Granovetter's threshold model of collective behaviour (Wiedermann et al., 2020; Granovetter, 1978). The model explains collective action from cascading contagious dynamics (Watts, 2002) of social activation in a complex social

network, exploring the interrelation of individual behavioral dynamics and network structures. In alignment with resource mobilization theory it assumes that a considered population can be divided into three distinct groups (Jenkins, 1983; McCarthy and Zald, 1977): (i) A small fixed population share $a$ consisting of *certainly active* individuals, such as instigators of a climate action, that deliberately act upon a certain issue. (ii) A fixed population share $c$ consisting of *contingently active individuals* whose opinions and norms align with the behaviour of the instigators but who did not join the climate action yet. (iii) A

remaining population share consisting of *certainly inactive* individuals who, due to opposing opinions and values, will never join a respective climate action. The first two groups taken together then form the fixed *potentially active population* share $p = c+a$. Actors in those groups are connected along ties in a social network with an average number of such ties per individual, $K$, commonly referred to as the 'average degree' of the social network (Newman, 2018). Actors become active via cascading dynamics if at least a fixed share $\rho$ of their neighbors is active as well (Watts, 2002). The model's dynamics can be described

by a one-dimensional discrete-time difference equation that computes the share of acting individuals, $r(t+1)$, at a given time $t+1$ as

$$r(t+1) = a + (p-a)F(r(t)). \tag{1}$$

Here, $a$ denotes the certainly active population share and $p$ the potentially active population share, i.e. those that can be mobilized via social activation. $F$ is the cumulative distribution function of the actors' activation thresholds, refereed to as

the emergent macroscopic threshold function. In other words, $F(r(t))$ represents the fraction of the contingent population (with relative size $p-a$) that take part in a given climate action once that movement has reached a relative size of $r(t)$. It has been first suggested (Granovetter, 1978) and then shown (Wiedermann et al., 2020) that such a threshold distribution $F$ with typically assumed properties, such as being broad-shaped and similar to a normal distribution, emerges from pairwise social interactions between individuals in their underlying social network. In particularly, assuming a random network topology given

by the Erdős–Rényi model as a first approximation for the unknown underlying social network (Erdős and Rényi, 1960) yields



the following analytical expression for $F$ (Wiedermann et al., 2020):

$$F(r) = 1 - \exp(-K) \sum_{b_i=0}^{\infty} \frac{(K - Kr)^{b_i}}{b_i!} \sum_{a_i=0}^{\left\lfloor \frac{\varrho b_i}{1-\varrho} \right\rfloor} \frac{(Kr)^{a_i}}{a_i!} \tag{2}$$

Note that, the emergent threshold function ($F(r(t))$) integrates information from individual behavioral thresholds and network structures and acts as a parsimonious tool for modelling social tipping processes, but, the threshold function itself is rather an outcome of these processes, and should not be treated as a social construct with causal effect in its own right. Also note that even though the approximation assumes the topology of an Erdős–Rényi random network, the emergent threshold function $F$ also represents other, less trivial, network structures sufficiently well (see below for details and SI for numerical simulations).

The fixed points $r^*$ of (1) can be estimated by setting

$$r^* = a + (p - a)F(r^*) \tag{3}$$

and solving for $r^*$ numerically. The model shows two saddle-node bifurcations with respect to both $a$ and $p$ (Wiedermann et al., 2020), which is the typical form of stability landscape for a tipping element, Fig. 3 (a). For sufficiently large $p$ and small values of $a$, i.e. $a = 0$ in the extreme case, the model displays an uncritical regime where no tipped state exists, Fig. 3 (a). Increasing $a$ pushes the system closer towards a bi-stable critical regime in which external influences, such as shocks or events, can lift the system onto an alternative stable state (yellow area in Fig. 3 (a)). Further increasing $a$ eventually triggers the commonly studied form of social tipping, i.e. bifurcation-induced tipping, where the untipped fixed point $r^*$ vanishes and the system enters the tipped regime (green area in Fig. 3 (a)). Notably, similar dynamics and a distinction into three qualitative regimes can be observed for increasing $p$ from low to large values, thereby giving rise to a cusp catastrophe (Wiedermann et al., 2020). In particular, for each value of $p$ there is a critical value of $a$ (given by the boundary between the red and yellow shaded area in Fig. 3) at which the system first enters the bi-stable critical regime. In that sense, increasing values of $p$ move the boundary between the uncritical and critical regime closer towards smaller values of $a$.

Given that the social network structure within the 66 countries that are included in our simulation are likely largely heterogeneous, and in many cases undetermined, we adopt a Monte-Carlo approach to check for the robustness of our results, numerically estimating the potential for social tipping and resultant instantiations of emergent threshold functions by randomly sampling unknown parameter values for $K$ and $\rho$. We observed that the resulting ensemble of threshold functions ($F(r)$) then not only resembles simple network topologies such as the Erdős–Rényi random network, but also sufficiently covers other, more realistic, network topologies. In particular, we perform robustness checks using an array of topologies (real-world empirical data about Facebook friendships, Barabási–Albert (Barabási and Albert, 1999), Watts–Strogatz (Watts and Strogatz, 1998), ring topologies, and random geometric networks) (Dall and Christensen, 2002), finding that in most cases, the ensemble threshold function fits estimates emerging from micro-simulation models, except in cases where the certainly active nodes are heavily clustered in modular networks (See Figs. A1 and A2 in Supplementary Information). This is to be expected, since in networks where the certainly active population is clustered within a highly modularized network structure, it is unlikely for the network to exhibit cascading processes resulting in social tipping across a broader population, as tipping would be contained





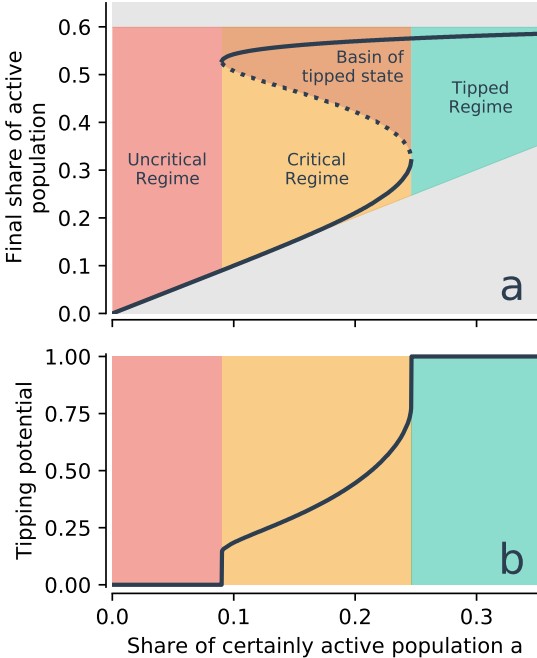

**Figure 3. a)** Exemplary bifurcation diagram of the social tipping model w.r.t. one of several influencing factors: the certainly active population share $a$. For low shares $a$ of certainly active individuals the system is in the uncritical regime (red shaded area) where no tipping is possible since only the lower branch exists. With increasing $a$ the system enters the critical regime (yellow shaded area) and event-induced tipping becomes possible if large enough shocks kick the steady state into the tipped state's basin of attraction (orange shaded area). With further increasing $a$ the model reaches its critical threshold for bifurcation-induced tipping above which only the upper branch exists (green shaded area). **b)** The tipping potential computed as the relative size of the basin of attraction of the upper branch.

to specific clusters and not through the network as a whole. However, we suggest that in the case of anticipation of SLR, real-world social networks are less likely to have such highly modularized network structures as for many countries, SLR affects broad sections of coastlines, stretching across diverse social and geographic groupings.

## 3.2 Estimation of certainly active population from SLR

We estimate the certainly active population share $a(T)$ per country by the proportion of individuals that are projected to be affected by sea-level rise (SLR) at a given *anticipation time horizon $T$* after 2020. Here, we assume individuals to be affected if they live at an elevation $X$ that lies at or below projected SLR at time $T$. Particularly, we use median SLR projections until 2300 obtained from the MAGICC climate model v2.0 (Nauels et al., 2017) for the RCP8.5 scenario that provides an upper bound of the RCPs (Riahi et al., 2011) (solid red line in Fig. 1(a)). Generally, the MAGICC climate model emulator estimates approximately 1m sea-level change for low-emission scenario RCP2.6 ranging up to approximately 5m sea-level rise for high-emission scenario RCP8.5 by the year 2300 (Fig. 1a). These projections are consistent with process-based models in IPCC





AR6 (Masson-Delmotte et al., 2021), which further note that even more severe SLR of more than 15m by 2300 cannot be
excluded under high emission scenarios (Portner et al., 2019).

In order to estimate the population-elevation distribution (Fig. 4) we combine country-specific gridded population data from
the Socioeconomic Data and Applications Center (SEDAC)  (Center for International Earth Science Information Network,
2016) with SRTM30 near-global digital elevation data (Farr et al., 2007), both provided at an angular resolution of $1/120°$.
Since entries in the SRTM30-data are truncated to full meters, we add uniformly distributed random noise of magnitude 1
meter to the entry of each grid cell. The median SLR projections are then combined with the country-specific population-
elevation distributions to obtain a time series of population shares that are affected by SLR until 2300. Software-packages to
parse and process both the gridded population and elevation data are available online at https://zenodo.org/record/4268015 and
https://github.com/marcwie/srtm30-parser.

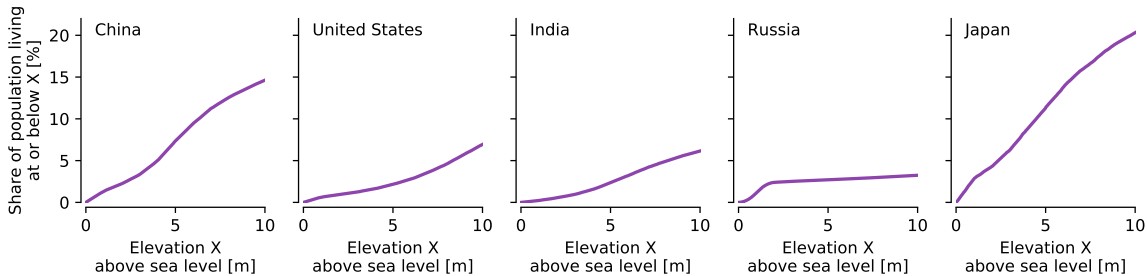

**Figure 4. Cumulative distributions of country-specific population shares living at or below an elevation level of $X$ above sea-level for
the five largest emitters of greenhouse gas that are considered in this study.**

### 3.3   Estimation of potentially active population from social survey data.

We estimate the potentially active population share $p_C$ in a country $C$ from subjects' expressed concern for climate change
in six different recent cross-national social survey programs: International Public Opinion on Climate Change (2022), Inter-
national Social Survey Programme (2010, Environment III), European Social Survey (2016, Wave 8), PEW Global Attitudes
Survey (Spring 2015), Eurobarometer (2017, EB 87.1; 2019, EB 91.3), and Life in Transition Survey (2010, II), see Tab. S1.
We assume that subjects that are not concerned at all are not potentially active. In contrast, subjects with the highest level of
concern are counted as surely belonging to the potentially active population share. For subjects with intermediate levels of con-
cern we assume a certain share of them is potentially active, using the following approach to estimate this share: Each survey $s$
contains an item capturing individual perceptions of climate change concern across multiple countries $C$. These items are given
on an ordinal scale, with varying numbers of outcomes $n_s$. We rescale those outcomes to take integer values $i_s = 0, 1, \ldots, n_s - 1$
such that $p_{i,C,s}$ gives the relative frequency of response $i$ in country $C$ and survey program $s$. We then estimate the potentially
active population share $p_{C,s}$ in country $C$ as

$$p_{C,s} = \frac{1}{n_s} \sum_{n_s} p_{i,C,s} \frac{i}{n_s - 1} \in [0, 1], \tag{4}$$





such that $p_{C,s} = 0$ if all participants were to respond $i_s = 0$ and $p_{C,s} = 1$ if all participants were to respond $i_s = n_s - 1$. In

total, we compile data for 66 countries with access to the sea. Some countries, especially in the European Union, are covered

by all six surveys, while other countries only appear in a single instance with an average coverage of 2.36 survey programs

per country. In cases where multiple survey programs available for a particular country, we then adopt the median value of

concern over all survey programs for that country as the estimated potentially active population share $p_C$ in Fig. 5 and Fig. 6.

An overview for estimated levels of concern across countries is given in Tab. S2 and Tab. S3.

### 3.4   Preprocessing, model setup and ensemble simulations

In alignment with earlier works (Marzeion and Levermann, 2014), we exclude countries with exceptionally low elevation

profiles, i.e, The Netherlands, Azerbaijan and Kazakhstan, that for large parts show elevation values even below present sea-

level. We additionally exclude all countries that are not adjacent to any larger body of water and, thus, exclusively lie inland,

e.g. Mongolia or Austria. In total, we excluded 22 countries from these analyses.

In a first step we vary the potentially active population share $p$ across its entire valid value range $p \in [0,1]$ in order to obtain

comprehensive statistics about the model's dynamics. We draw $N = 2000$ random combinations of parameters $\rho \in (0,1)$ and

$K \in (0,100)$ and compute $r^*$ numerically from Eq. (3) for every combination of $p \in [0,1]$ and $a(T)$, for $T = 0, 1, \ldots, 280$. In

a second analysis, we then fix the potentially active population share in each country, $p_C$, according to the estimated levels of

concern (see above).

### 3.5   Tipping potential and intervention sizes.

The tipping potential in Fig. 5 and Fig. 6 is computed as the average basin stability (Menck et al., 2013) of the tipped fixed

point $r_u^*$ (subscript $u$ for *upper* branch) in Fig. 3 taken over an ensemble of $N = 2000$ simulations with randomly drawn values

of $K$ and $\rho$ (see above). For a single simulation $s$ we define the tipping potential $u_{tot}(s)$ as

$$
u_{tot}(s) = \begin{cases} 0 & \text{if only untipped lower branch exists} \\ \frac{p - r_m^*}{p - a} & \text{if unstable middle branch exists} \\ 1 & \text{if only tipped upper branch exists} \end{cases}
\tag{5}
$$

Here, $r_m^*$ (subscript $m$ for *middle branch*) is the location of the fixed point corresponding to the unstable branch of the model's

bifurcation diagram (dashed line in Fig. 3(a)), $p$ is the potentially active population share and $a$ is the certainly active population

share. The average tipping potential $u_{tot}$ is then given as

$$
u_{tot} = \langle u_{tot}(s) \rangle_s
\tag{6}
$$

$$
= \frac{1}{N} \sum_{s \mid u_{tot}(s) \in (0,1)} u_{tot}(s) + \frac{1}{N} \sum_{s \mid u_{tot}(s) = 1} 1
\tag{7}
$$

$$
= \frac{1}{N} \sum_{s \mid u_{tot}(s) \in (0,1)} u_{tot}(s) + u_{bif}.
\tag{8}
$$





The average tipping potential $u_{tot}$ measures the combined effects of event-induced (first term in the rhs. of Eq. (8)) and bifurcation-induced tipping (second term in the rhs. of Eq. (8)) Each contribution to $u_{tot}$ is thus either $0 < u_{tot}(s) < 1$ if the system is in the critical regime and $u_{tot}(s) = 1$ if the system is in the tipped regime (Fig. 3(b)). When aggregated over all simulations $s$, the bifurcation-induced tipping potential $u_{bif}$ computes the share of simulations in which bifurcation-tipping occurred (depicted in Fig. 5 third row) and forms a direct contribution to $u_{tot}$.

For all simulations $s$ where the model's steady state falls into the critical regime (yellow shaded area in Fig. 3), we compute the required intervention size $I(s)$ to tip the equilibrated system from the stable untipped state $r_l^*$ (subscript $l$ for *lower* branch) into the stable tipped state. Specifically, we express $I(s)$ as the minimum contingent population share $c = p - a$ required to lift the system from $r_l^*$ *over* the middle branch $r_m^*$. This yields

$$I(s) = \begin{cases} \frac{r_m^* - r_l^*}{p-a} & \text{if } 0 < u_{tot}(s) < 1 \\ 0 & \text{otherwise} \end{cases} \tag{9}$$

The required intervention sizes $I$ depicted in Fig. 5 are then computed as the respective percentiles of the set of all ensemble members $s$ for which we obtain positive values of $I(s)$.

## 4 Results

### 4.1 Anticipation and concern foster complementary forms of social tipping

The tipping potentials are first estimated for the five countries with the largest greenhouse gas emissions (China, US, India, Russia and Japan) (iea, 2020) for varying anticipation time horizons (e.g. 100, 150 or 200 years into the future) and potentially active population shares. We then further assess the corresponding tipping potentials for the fixed potentially active population share according to the estimated level of climate change concern in each country (see also Sect. 3).

We find that the overall tipping potential increases strongly with greater potentially active population shares. A notable dependency of this tipping potential on the time horizon can especially be observed for countries where a larger part of the population lives at or close to sea-level, as in the case of Japan or China (Fig. 5 (a)).

With extended anticipation time horizons, the bifurcation-induced tipping potential, i.e. those cases where the system *shifts* into the tipped regime, increases most significantly if the potentially active population is larger than at least approximately 20% (see Fig. 5 (b)) – a level that is surpassed in all five considered countries. For the estimated values of the potentially active population, our model suggests increased bifurcation-induced tipping potential in countries with large near-sea-level population density, particularly China and Japan (Fig. 5 (c)). This implies that while the overall tipping potential is relatively similar across contexts, bifurcation-induced tipping becomes more likely in regions which are more vulnerable to future SLR.

However, compared with the overall large potential, bifurcation-induced tipping remains comparatively unlikely (cf. Fig. 5 (a) and (b)). This implies that even though an alternative stable state of collective climate action may exist, the system is unlikely to reach this state by mere crossing of a single critical threshold or tipping point. This is because in most cases, the share of certainly active population directly affected by SLR is not sufficient to trigger such bifurcation-induced tipping.





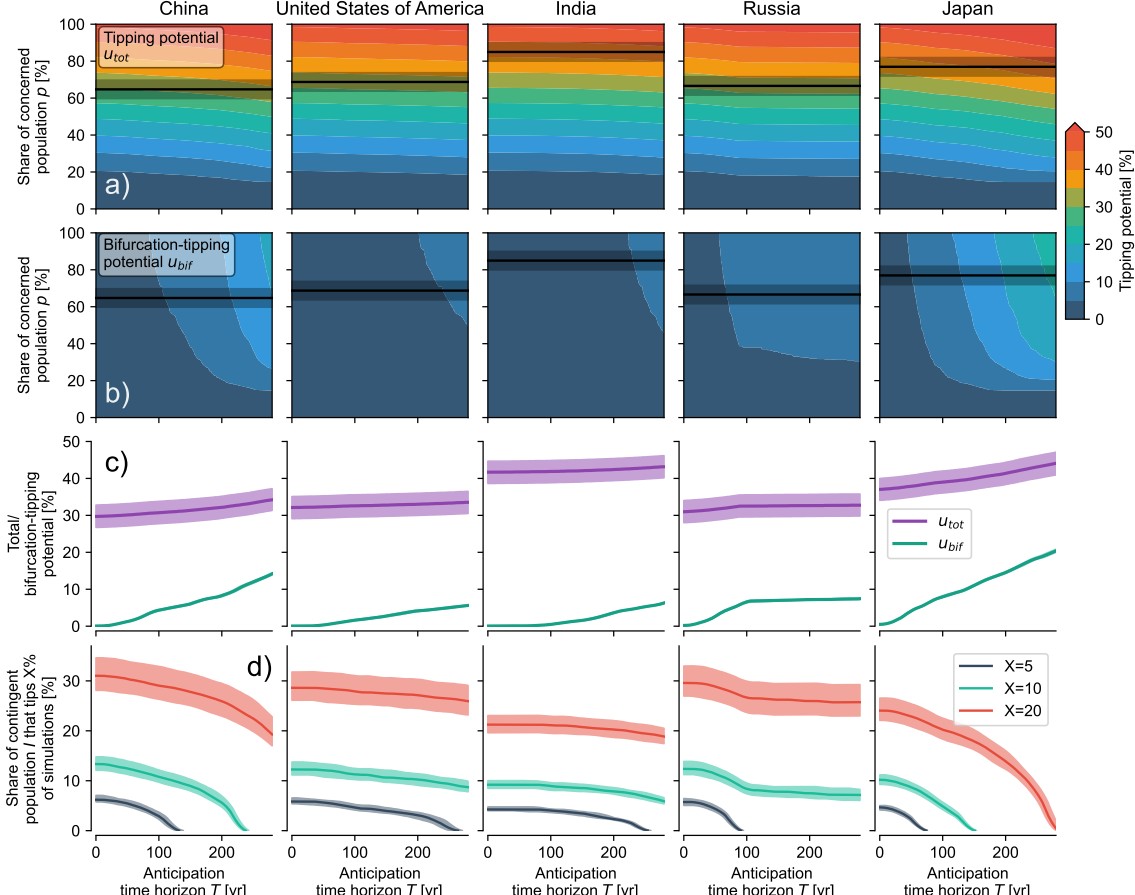

**Figure 5. Social tipping potential for the five highest-greenhouse gas emitting countries China, USA, India, Russia and Japan (in terms of CO2 equivalents). (a)** Total tipping potential and **(b)** bifurcation-induced tipping potential for varying shares of concerned population $p$ and anticipation time horizon $T$, from zero (blue) to more than 50% (red). Black lines indicate the estimated country-specific climate change concern $p_C$ with a shaded interval of $\pm 5\%$ deviation, indicating that variations in the level of concern do not qualitatively alter the results. **(c)** Total, $u_{tot}$, and bifurcation-induced tipping potential, $u_{bif}$, for the estimated levels of country-specific climate change concern. Shaded areas indicate variations of $u_{tot}$ and $u_{bif}$ within the $\pm 5\%$-band of concern. **(d)** Size $I$ of necessary interventions to tip a given share $X$ of simulations once a country's social system reaches the critical regime. The temporal axis reflects the anticipation time horizon of $T$ years prior to 2020, i.e. $T = 0$ implies that individuals do not anticipate any future impacts while the maximum possible value, $T = 280$, assumes individuals to anticipate sea-level impacts up to 2300, the penultimate year in the MAGICC projections (see also Fig. 1).

300  Instead, the system is mostly found in a critical, yet not tipped, state where interventions then carry the potential to kick the system into its alternative stable state via event-induced tipping. The required intervention size, measured in terms of the contingent population share, generally decreases as the anticipation time horizon grows, moving the system closer towards the tipped state (see Fig. 3). Comparatively small intervention sizes (less than 10%) of spontaneously activated populations can



suffie to trigger transitions to the alternative state with 5% chance, indicating a potential for event-induced tipping (see black
lines in Fig. 5 (d)).

Additionally, our model indicates that countries with larger potentially active population shares have correspondingly lower
required intervention sizes for initiating event-induced tipping (as for instance India and Japan, see Fig. 5 (d)), as the gap is
lowered between an untipped stable state and the basin of attraction of the tipped state (cf. last row of Fig. 5 and Fig. 3). But,
even in cases of the highest climate change concern, such as in India, our model indicates that a non-zero intervention size is
needed when the anticipation time horizon is shorter, implying that at least some intervention is necessary to trigger tipping
processes. Increasing the anticipation time horizon translates into substantially smaller required intervention sizes, eventually
even reaching zero for the modeled cases of Japan and, in parts, China (Fig. 5 (d)).

## 4.2 Three global classes of tipping processes

Drawing upon a broader comparative analysis covering 66 countries with access to the sea, we find that concern for climate
change and the total tipping potential are strongly correlated (Fig. 6 (a) and (b)). Accordingly, we identify three qualitative
classes of tipping processes facilitating climate action that vary by the country-specific values of total and bifurcation-induced
tipping potentials (Tab. 1).

**Tipping Class I – Large total and high bifurcation-induced tipping potential:**

In these countries, the emergence of social tipping processes is fostered by heightened levels of climate change concern and
SLR-sensitive elevation-population profiles. In this case, an increase in the anticipation time horizon or a relatively small
intervention size can trigger a transition to an alternative state of increased climate action. Therefore, such social systems can
be considered to already be in a rather critical state.

Examples for this Class I primarily include countries along the Pacific Ocean, such as Indonesia, Japan, Vietnam, as well as
China. Here, SLR is likely to affect large population shares due to large metropolitan areas located near the coast (Kulp and
Strauss, 2019). Combined with overall heightened climate change concern in these countries (cf. Fig. 6 (a)), individual events
and expanded anticipation time horizons both have the potential to instigate social tipping processes in our model (Tab. 1).

**Tipping Class II – Large total and low bifurcation-induced tipping potential:**

Countries in this class are characterized by high levels of concern and low population shares likely to be affected by projected
SLR. This indicates that the corresponding countries might be sufficiently close to or already in a critical state, where rela-
tively small interventions would suffice to induce transitions towards an alternative state via event-induced tipping. However,
bifurcation-induced tipping pathways resulting from expanded anticipation time horizons are comparatively unlikely due to
less SLR-sensitive elevation-population profiles.

Class II mostly covers countries in South America, Africa and along the Indian Ocean. Climate change concern in those
countries can often be attributed to more short-term impacts than those related to SLR, such as water scarcity, more frequent





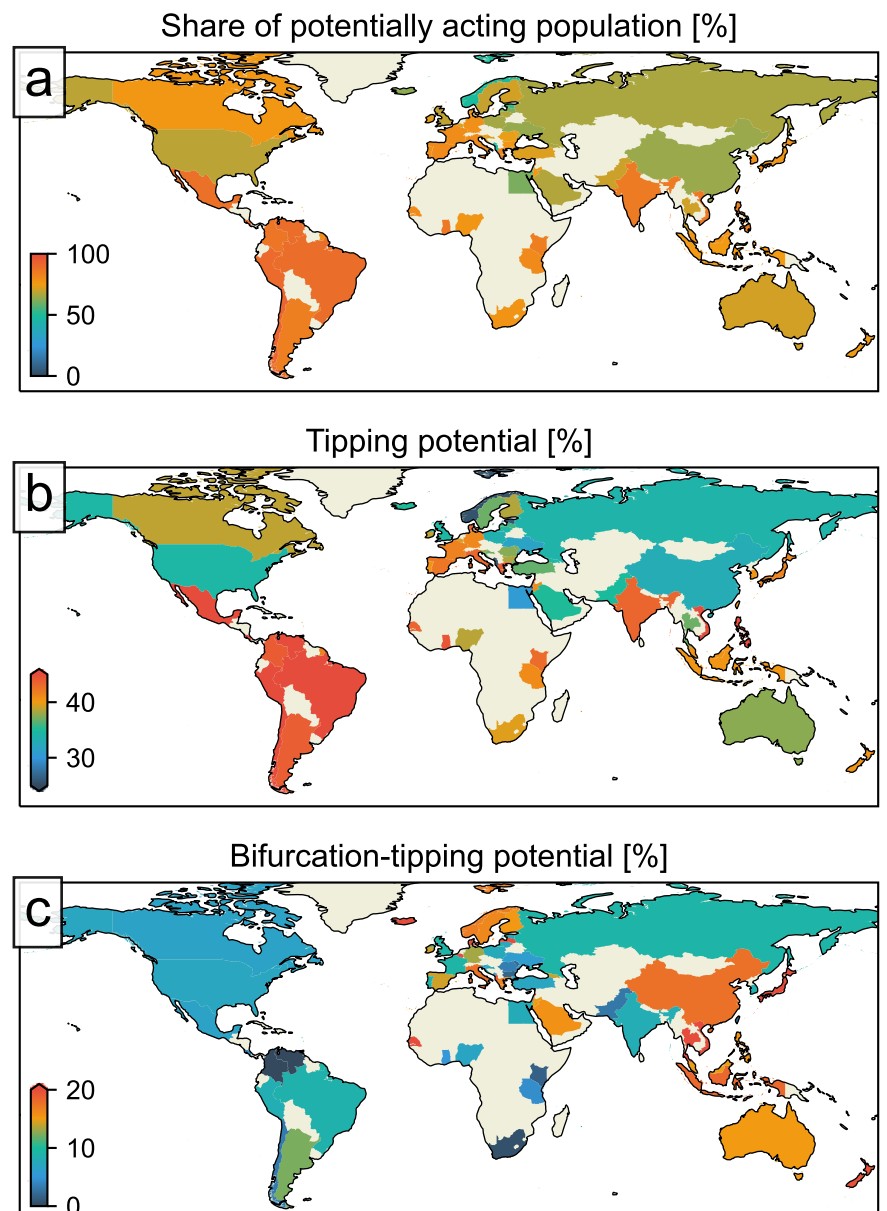

**Figure 6. Cross-national potential for social tipping towards increased climate action. (a)** Estimated country-specific potentially active population share. (Total) tipping potential **(b)** and bifurcation-induced tipping potential **(c)** for the estimated values of climate change concern from social survey data and the largest possible anticipation time horizons. Countries are geographically clustered according to their tipping classes I-III, i.e. countries with large total and bifurcation-induced tipping potential (Tipping Class *I*) in the Indo-Pacific, countries with large total and low bifurcation-induced tipping potential (Tipping Class *II*) in the remaining Southern Hemisphere, and low total and bifurcation-induced tipping potential (Tipping Class *III*) in Europe and North America.



**Table 1. Countries tend to cluster regionally with respect to their total and bifurcation tipping potentials in response to anticipated sea level rise.**

|  | Class I | Class II | Class III |
|---|---|---|---|
| Total Tipping Potential | Large | Large | Small |
| Bifurcation Tipping Potential | High | Low | Low |
| SLR Exposure/Anticipation | Greater | Lower | Lower |
| Climate Change Concern | High | High | Lower |
| Regional Clusters | West Pacific rim countries, China, Indonesia, Japan, Vietnam | South America, Africa, Indian Ocean rim | North America, Europe |

extreme weather events, or shifts in precipitation variability (Howe et al., 2019). This is also reflected within the scope of our model, as the total tipping potential (driven by climate change concern) and the bifurcation-induced potential (driven by anticipated SLR impacts) are largely uncorrelated (Fig. 6 (b) and (c)). We suggest that such temporally more immediate climate impacts could potentially trigger a rapid short-term mobilization of a large population share, thereby acting as a (non-deliberate) social tipping intervention, which can effectively kick the respective social systems into an alternative state of increased climate

action via event-induced tipping processes.

**Tipping Class III – Small total and small bifurcation-induced tipping potential:**

This class is exemplified by cases where both the elevation profile and low climate change concern effectively hinder social tipping. Accordingly, only large intervention sizes potentially push the system towards an alternative state of increased climate action. Further increasing concern could lower the baseline of the required intervention sizes, thereby rendering the system

more critical (moving it towards Tipping Class II). However, due to the country-specific elevation profiles, increased anticipation time horizons are expected to only have minor influences on the intervention sizes required for triggering social tipping, as the projected impacts of SLR remain comparatively small.

Class III covers North America and parts of Europe. Here, immediate threats from SLR are comparatively small since large population shares live further inland. Likewise, climate change concern is comparatively lower (cf. Fig. 6(a)), potentially due

to larger perceived psychological distances to these risks (Spence et al., 2012). Such psychological distances can result from comparatively higher economic development, larger perceived preparedness for the impacts of climate change (Spence et al., 2012) and decreased vulnerability (Maiella et al., 2020). Thus, according to our model, social tipping in these counties is fostered largely through elevating climate change concern (Singh et al., 2017). Then, once concern reaches a sufficiently high level, other events or interventions would have the potential to kick the system into an alternative state of increased climate

action.

Notably, a fourth tipping class with low total tipping potential and a large bifurcation-induced tipping potential cannot exist since by definition the former exceeds the latter (see Sect. 3).





## 5 Conclusions

### 5.1 Summary and discussion

Here, we adopt a model for social tipping to explore transformative pathways towards an alternative state of increased climate action resultant from climate change concern and anticipated impacts of projected sea-level rise across 66 countries with access to the sea. We find that in many countries, climate change concern has reached a point where the system can be considered critical, such that an alternative stable tipped state exists (where a large population share engages in climate actions), and sensitive interventions (e.g. policy regime changes or social movements) or stochastic events (e.g. storm surges or floods) have

the potential to push the system towards that tipped state.

Our model suggests that climate change concern has reached sufficiently high levels in many countries such that social tipping processes towards increased climate action could be instigated. However, concern alone is insufficient to trigger such social tipping processes. Rather, concern functions to create *critical*, enabling conditions under which a system becomes more susceptible to social tipping interventions (such as by political or civil society actors). Expanded anticipation time horizons can

directly lower the required intervention size, thereby increasing the likelihood for social tipping dynamics to emerge.

As such, the two factors, climate change concern and anticipation time horizons, are of a complementary yet mutually beneficial nature. Increases in concern make tipping dynamics more likely to occur following external influences or interventions, while the more *deterministic* process largely manifests through increased anticipation time horizons. As the anticipation time horizons necessary to trigger tipping are often on a centennial timescale, event-induced tipping (Ashwin et al., 2012) via

interventions presents a comparatively more likely pathway towards instigating social tipping processes in the near future.

In this way, in our model social tipping processes largely differ from those in climate systems (see also (Winkelmann et al., 2022)), which often focus on identifying scenarios where a (single) control parameter crosses a critical threshold, e.g., in global mean surface temperature, as the instigator of systemic transgression into an alternative stable state (Lenton et al., 2008). Instead, the notion of sensitive intervention (Farmer et al., 2019) and event-induced tipping becomes ever more important for

social tipping processes, as critical thresholds of control parameters might be infeasible to reach or impossible to predict.

A key finding from our modeled results is the emergence of three qualitative classes of social tipping. Here we identify a transformation pathway towards increased potential for social tipping towards climate action. While the tipping potential within a Class III-state is comparatively low it increases with climate change concern such that the possibility for tipping processes begins to exist (i.e. leading to a Class II-state). Then, extended anticipation time horizons reduce the intervention

sizes necessary to trigger tipping processes (Tipping Class I), such that a comparatively smaller kick within the system or it's environment carries the potential to instigate social tipping processes.

Given the increasing social relevance of anthropogenic climate change, identifying social systems that are in more critical states can guide strategic policy entrepreneurs (Kingdon, 1995). Targeted interventions can amplify critical conditions or can even provide a substantial enough kick to instigate climate actions. For example, climate change social movements can play

a crucial role, providing a mechanism to develop new political coalitions (Weible and Sabatier, 2017), thereby increasing the salience of climate change as a political issue. Such increased salience can effectively open a policy window, allowing climate





policy advocates to promote their agendas (Kingdon, 1995). Further large shift in public opinion can punctuate sticky environmental regulatory regimes (Baumgartner and Jones, 2010) and political-institutional lock-ins (Cecere et al., 2014), resulting in the adoption of new public policies, such as carbon taxation (Lemoine and Traeger, 2016) or incentivizing development of
alternative energy supplies (Patt and Lilliestam, 2018).

In contrast to common behavioral contagion modeling approaches (Dodds and Watts, 2004b; Böttcher et al., 2017), our country-specific estimations of social tipping potentials are uniquely based on real-world observations (i.e., projected SLR impacts, measured elevation-population distributions and a comprehensive review of social survey data on climate change concern). We thereby contribute a novel perspective to the growing field of social-ecological systems modeling that aims
to specifically account for the dynamic interactions between biogeophysical Earth and social systems on up to planetary scales (Müller-Hansen et al., 2017; Beckage et al., 2018; Donges et al., 2020; Beckage et al., 2022; Moore et al., 2022).

Additionally, our modeling exercise expands upon previous studies of threshold dynamics and social tipping that have largely focused on social systems in isolation, either theoretically (Granovetter, 1978; Schelling, 1971), in controlled laboratory experiments (Centola et al., 2018; Centola and Baronchelli, 2015) or via network-based numerical simulations (Watts, 2002;
Singh et al., 2013; Karsai et al., 2016).

Overall, we find that concern for climate change increases the criticality and likelihood for social tipping processes, and expanded anticipation time horizons and growing shares of active population move the system closer towards a critical threshold, effectively reducing the required intervention size needed to ultimately kick the system into a alternative state where climate action becomes the social norm. Our study highlights the potential that lies in the combination of these factors, pushing social
systems closer towards positive tipping points, and is thus a first step towards truly closing the loop from climate impacts to substantive societal transformation (Donges et al., 2017).

## 5.2  Limitations

We acknowledge that social tipping processes can be triggered by a variety of distinct external factors such as in the aftermath of extreme events (Demski et al., 2017; Konisky et al., 2016) or in response to large-scale environmental changes (Ioris et al.,
2020). However, our model simulates the potential for social tipping solely resultant from anticipation of projected SLR. We find that, on its own, SLR is comparatively unlikely to result in bifurcation-induced tipping. However, our approach does not yet account for other potential impacts that are directly related to SLR, such as increased flood risks, coastal erosion, or increased vulnerability to storm surges (Kulp and Strauss, 2019). These impacts are comparatively more likely to affect even larger population shares in coastal areas, and are likely to occur in coming years and decades (Muis et al., 2016; Buchanan et al.,
2016; Tebaldi et al., 2012). As such, our modeling approach can be considered rather conservative in estimating potentials for tipping processes triggered by SLR-related impacts.

For now, we assume a straightforward conversion of SLR into certainly active populations as a probable scenario of subjective risk assessment, in which individuals simply compare official projections with their knowledge of the elevation profile of their local region. However, SLR is projected to vary regionally, with coastal areas around the Indian Ocean and Western



Pacific being affected more severely than other locales (Perrette et al., 2013). Since these are also regions where climate change concern is highest, such regional SLR differences could result in even greater potential for social tipping.

Observation-based research have noted how social contagion varies by geographic factors, such as the physical distance an actor has to travel to participate (Traag et al., 2017), individual propensity for behavioral change (State and Adamic, 2015), as well as the complexity in contagion spreading across social network structures (Guilbeault and Centola, 2021) and forms of

social interaction (Iacopini et al., 2019). Given the potentially heterogeneous social network structures across the 66 countries, we assessed the robustness of our results using numerical Monte Carlo simulations of cascading dynamics across a diverse set of common network topologies. We found that in most cases, there is a parametrization of the emergent threshold function $F$ that is similar to what one would expect from certain other, non-trivial, network topologies. In that sense, our modeling approach can not only be interpreted as an ensemble of different Erdős–Rény networks, but an ensemble of different network

topologies itself (see Supplementary Materials for detailed further discussion).

We ultimately note that, while illustrative, neither the mechanism we explore here nor the underlying model results should be interpreted in an overly-deterministic manner. For example, tipping can in fact occur in any of the three identified classes, as concern is large enough for most countries so that they might have already entered into a critical regime. Rather, our findings suggest that as countries transition towards Tipping Class I, the tipping potential increases and the intervention size necessary

to trigger social tipping decreases correspondingly.

## 5.3 Outlook

In addition to natural drivers, deliberate interventions by policy makers and climate actors carry the potential to instigate social tipping processes (Lenton, 2020), for instance via financial disclosure (Farmer et al., 2019), information feedbacks (Otto et al., 2020a), or climate movements (McAdam, 2017). To assess the potential for deliberately initiated societal transformations, i.e.

positive social tipping (Tàbara et al., 2018; Hinkel et al., 2020; Lenton et al., 2022; Fesenfeld et al., 2022), future research should focus on identifying processes and mechanisms that lower the intervention sizes needed to kick the system into, or increase the potential to shift the system closer towards, an alternative state of increased climate action. Thus, the actual type of events instigating social tipping processes can emerge from an array of sources (e.g. natural or social) across varying scales (e.g. macro or micro). For now, our model is agnostic to the specific form of the instigating event, but rather aims to identify

factors affecting the tipping potential of systems.

The individual likelihood to engage in climate actions results from an interplay of internal (i.e. concern, values, attitudes and beliefs) and external (contextual, political, economic) factors, as well as a multitude of potential barriers (i.e. current behaviors, adaptive capacity, social norms) (Kollmuss and Agyeman, 2002). Our proposed model currently assumes that behavioral change is not associated with costs to the individual. Moreover, there are unique factors within each context, many of which

may even be unobservable or unknowable, that affect the capacity for tipping processes in a given social system. Still, our idealized scenario shows that due to the inherent complexity of social systems (Schill et al., 2019; Levin et al., 2013) tipping processes arise from multiple intertwined and mutually co-dependent factors which potentially lead to alternative future trajectories, for example in energy, financial, or socio-political systems (Otto et al., 2020a; Farmer et al., 2019; Lenton, 2020; Sharpe



and Lenton, 2021; Tàbara et al., 2018). Future research should therefore factor in relevant heterogeneities within and between
countries, such as different forms of agency (Otto et al., 2020b; Stadelmann-Steffen et al., 2021), shifting social norms (Nyborg
et al., 2016), cultural dimensions (Hofstede and Bond, 1984), the susceptibility of the country (or region) to a broader range of
climate impacts, or the ability to participate in social movements or civil society.

We explore one mechanism for social tipping processes, noting a potential transformative pathway resulting from the interre-
lation of climate change concern and anticipation of SLR via adoption of a social activation model. Given the necessity for rapid
societal transformations (Rockström et al., 2023), and the potential for social tipping processes to instigate such changes (Otto
et al., 2020a; Winkelmann et al., 2022), we encourage further research in this field. Future work can explore global condi-
tions instigating social tipping, highlight unique mechanisms within country or region-specific case studies, explore the role of
short-term shocks (such as extreme events or rapid political mobilization) as triggers (Ricke and Caldeira, 2014), and identify
the distinct circumstances under which diverse social tipping elements are likely to experience transformative changes.

*Data availability.* The SRTM30 elevation data is available from the US Geological Survey.

The Gridded Population data of the Earth (GPW) v2 is obtained from the Socioeconomic Data and Applications Center (SEDAC). Software
packages for downloading and preprocessing of the data are https://github.com/marcwie/sedac-gpw-parseravailable here and here.

The MAGICC v2.0 sea-level rise projections are computed using the model source code available at and using supplementary input data
from here.

International Social Survey Programme (2020, "Environment IV") data available here.

International Public Opinion on Climate Change (2022) data are available here.

European Social Survey (2016, Wave 8) data are available here.

Eurobarometer (2017, 87.1) data are available here.

Eurobarometer (2019, 91.3) data are available available here.

Life in Transition Survey (2010, Wave II) data are available here.

Pew Global Attitudes Survey (Spring 2015) data are available here.

Survey data replication coding and data collection materials are available via author's Github GitHub.

## Appendix A: Robustness Checks of Macroscopic Approximation

Within this study, we utilize a macroscropic approximation of the threshold function, $F(r(t))$, representing the fraction of
the contingent population (with relative size $p - a$) that takes part in a climate action. This approximation is derived analyti-
cally (Wiedermann et al., 2020), assuming a Erdős–Rényi model for the unknown underlying social network (Erdős and Rényi,
1960).

Given that the true social network structure in any of the 66 countries simulated in this modeling exercise is unknown, we
follow a logic akin to bayesian non-informative priors and use a Monte Carlo simulation approach, choosing a random average
degree ($K$) and threshold value ($\rho$) for the macroscopic approximation, which results in an ensemble of sigmoid-shaped curves





for the emergent threshold distribution ($F$) varying broadly in both the location and steepness of their inflection points. This approach thereby generates an ensemble of different shapes of $F$ utilized in our simulations, including for example: (i) a step function (for large $K$ and $\rho = 1$), (ii) an S-shaped curve (for intermediate $K$ and $\rho$), (iii) a monotonic increase above the main diagonal for small $\rho$, and (iv) a monotonic increase below the main diagonal for large $\rho$. Hence different parameterizations are

already considered when computing the average tipping potentials displayed in Figs. 3 and 4 of the paper. In that way, our approach is conservative in that it integrates widely across even qualitatively distinct forms of threshold functions as we do not make strong assumptions about any specific such form.

Notably, the Erdős–Rényi model is comparatively parsimonious and may not well represent more highly clustered network structures (Centola et al., 2018; Guilbeault and Centola, 2021). Accordingly, we engaged a series of robustness checks, com-

paring how well the ensemble of threshold functions emergent from our Monte Carlo simulations cover microscopic network dynamics across a range of network topologies: Barabasi-Albert (BA) (Barabási and Albert, 1999), Watts-Strogatz (WS) with rewiring probability $\beta = 0.25$ (Watts and Strogatz, 1998), a ring topology (Watts-Strogatz with $\beta = 0$), a Random Geometric Network (RGG) (Dall and Christensen, 2002) and real-world data from Facebook ( 63k nodes, avg. degree 26).

In general, we find that for all random topologies (Fig. A1), the ensemble of macroscopic approximations covers the em-

pirical results from the above-mentioned additional micro-simulation models rather well when the certainly active nodes are sufficiently dispersed across the network. In other words, in most cases, there is a combination of $\rho$ and $K$ in the Erdős–Rényi network that produces an emergent threshold function $F$ that is similar to what one would expect from certain other network topologies. In that sense, our Monte-Carlo approach can not only be interpreted as an ensemble of different Erdős–Rény networks, but an ensemble of different network topologies itself.

But, of particular note, when the certainly active nodes are closely clustered, we are less likely to observe tipping-like processes exemplified by this macroscopic approximation (esp. lower panel of Fig. A2). When the certainly active population is clustered within a highly modularized network structure, it is unlikely for the network to exhibit cascading processes resulting in social tipping across a broader population, as tipping would be contained to specific clusters and not penetrate through the network as a whole.

We suggest that in the case of anticipation of SLR, real-world social networks are less likely to have such highly modularized network structures. For many countries, SLR affects broad sections of coastlines, stretching across diverse social and geographic groupings. Furthermore, the effects of SLR are unlikely to be only observed and experienced by those directly impacted, rather these are likely to spill-over to broader geographic regions and social groups (e.g. through climate induced migration, mass media coverage) even though these are not specifically considered in the present manuscript. In such cases,

we assume that a high clustering of the certainly active population within a modularized network structure is less likely to be representative of the actual network structure of the 66 countries simulated in this modeling exercise.

Further, for our research design, we explicitly chose a common level of complexity across all components. That is, the emphasis of these findings should not be too heavily on either the individual social, climate or network aspects, but rather the combined implementation of these factors. And this level of complexity is set at a lower-level to specifically allow for exploring

conceptual scenarios. By keeping the modeling components on a relatively simplified level, we aim to avoid the tendency of



assuming predictive capacity via the increased complexity of the modeling approach. In this case, we chose a macroscopic approximation of network topology that is comparatively simple, yet as we find, robust across a number of other potential structures.

*Author contributions.* R.W. and J.F.D. conceived the study. E.K.S. and M.W. designed the research, developed and analyzed the model
results, and co-wrote the manuscript. J.F.D., J.H. and R.W. contributed to the model design and analysis. All authors contributed to the writing of the manuscript.

*Competing interests.* At least one of the (co-)authors (Jonathan Donges) is a member of the editorial board of Earth System Dynamics.

*Acknowledgements.* We thank Alexander Nauels for providing the sea-level rise projections computed from the MAGICC climate model v2.0. We also thank Julia Bognar, Christina Eder and Alexia Katsanidou for their insightful feedback and suggestions. This work was devel-
oped in the context of the COPAN collaboration at the Potsdam Institute for Climate Impact Research (PIK). M.W. and K.S. are supported by the Leibniz Association (project DOMINOES). J.F.D. is grateful for financial support by the Stordalen Foundation via the Planetary Boundary Research Network (PB.net), the Earth League's EarthDoc program, the European Research Council advanced grant project ERA (Earth Resilience in the Anthropocene; grant ERC-2016-ADG-743080), and the German Federal Ministry for Education and Research (BMBF) via the project 'PIK Change' under grant 01LS2001A. The authors gratefully acknowledge the European Regional Development Fund (ERDF),
BMBF and the Land Brandenburg for providing resources on the high-performance computer system at PIK.





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





**Figure A1. Robustness checks of macroscopic approximation - random network topologies.** Each panel represents results from randomly chosen certainly active nodes across divergent network topologies. Each dot represents micro-simulations results, while the lines represent macroscopic approximations for nine exemplary combinations of the two parameters $\rho$ and $K$ – the actual ensemble contains a wider range of combinations.



**Figure A2. Robustness checks of macroscopic approximation - clustered, segmented and differential degrees in diverse network topologies.** Each panel represents results from clustered, segmented and varied degrees of certainly active nodes across divergent network topologies. Each dot represents micro-simulations results, while the lines represent macroscopic approximations for nine exemplary combinations of the two parameters threshold rho and average degree K – the actual ensemble contains a wider range of combinations.





| Data | Year | Question Wording | Item Coding | Number of Outcomes $n_s$ | Mean | Std. Dev. |
|---|---|---|---|---|---|---|
| Eurobarometer (EB 87.1 / EB 91.3) | 2017/ 2019 | And how serious a problem do you think climate change is at this moment? | 1 'Not at all serious' to 10 'Extremely serious' | 10 | 0.75 | 0.05 |
| ESS | 2016 | How worried about climate change? | 1 'Not worried' to 5 'Extremely worried' | 5 | 0.51 | 0.06 |
| ISSP | 2021 | Do you think that a rise in the world's temperature caused by climate change is dangerous for the environment? | 1 Not at a;; to 5 'Extremely dangerous' | 5 | 0.74 | 0.04 |
| IPOCC | 2022 | How worried are you about climate change? | 1 'Not at all worried' to 4 'Very worried' | 4 | 0.78 | 0.07 |
| LITSII | 2010 | How concerned are you about climate change? | 1 'Not concerned' to 5 'Extremely concerned' | 5 | 0.58 | 0.08 |
| PEW2015 | 2015 | In your view, is global climate change a problem? | 1 'Not a problem' to 4 'Very serious problem' | 4 | 0.78 | 0.09 |

**Table A1.** Social Survey Data Sources, Question Wording, Items and Descriptive Statistics

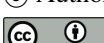



| Country | EB 87.1 2017 | EB 91.3 2019 | PEW Global Attitudes 2015 | ISSP 2021 | LITSII 2010 | ESS 2016 | Median |
|---|---|---|---|---|---|---|---|
| Albania | - | - | - | - | 0.49 | - | 0.49 |
| Argentina | - | - | 0.83 | 0.76 | - | - | 0.80 |
| Australia | - | - | 0.71 | 0.58 | - | - | 0.64 |
| Belgium | 0.72 | 0.73 | - | 0.60 | - | 0.55 | 0.66 |
| Brazil | - | - | 0.94 | - | - | - | 0.94 |
| Bulgaria | 0.77 | 0.80 | - | 0.74 | 0.63 | - | 0.75 |
| Canada | - | - | 0.76 | 0.68 | - | - | 0.72 |
| Chile | - | - | 0.92 | 0.83 | - | - | 0.88 |
| China | - | - | 0.65 | - | - | - | 0.65 |
| Croatia | 0.73 | 0.74 | - | 0.75 | 0.65 | - | 0.73 |
| Cyprus | 0.78 | 0.82 | - | - | - | - | 0.80 |
| Denmark | 0.77 | 0.80 | - | 0.62 | - | - | 0.77 |
| Estonia | 0.59 | 0.64 | - | - | 0.48 | 0.41 | 0.54 |
| Finland | 0.71 | 0.72 | - | 0.63 | - | 0.51 | 0.67 |
| France | 0.78 | 0.80 | 0.82 | 0.62 | 0.50 | 0.55 | 0.70 |
| Georgia | - | - | - | - | 0.67 | - | 0.67 |
| Germany | 0.75 | 0.79 | 0.79 | 0.73 | 0.52 | 0.59 | 0.74 |
| Ghana | - | - | 0.86 | - | - | - | 0.86 |
| Greece | 0.81 | 0.84 | - | - | - | - | 0.83 |
| Iceland | - | - | - | 0.57 | - | 0.53 | 0.55 |
| India | - | - | 0.91 | - | - | - | 0.91 |
| Indonesia | - | - | 0.72 | - | - | - | 0.72 |
| Ireland | 0.71 | 0.77 | - | - | - | 0.44 | 0.71 |
| Israel | - | - | 0.63 | 0.70 | - | 0.42 | 0.63 |
| Italy | 0.80 | 0.80 | 0.82 | - | 0.68 | 0.55 | 0.80 |
| Japan | - | - | 0.76 | 0.79 | - | - | 0.77 |
| Jordan | - | - | 0.76 | - | - | - | 0.76 |
| Kenya | - | - | 0.84 | - | - | - | 0.84 |
| Latvia | 0.62 | 0.66 | - | 0.59 | 0.51 | - | 0.60 |
| Lebanon | - | - | 0.85 | - | - | - | 0.85 |
| Lithuania | 0.74 | 0.73 | - | 0.67 | 0.62 | 0.47 | 0.67 |

**Table A2.** Estimated shares of potentially acting individuals from weighted averages over all responses in the six survey programs. Dashes indicate that a country is not covered by the specific survey program. Countries with initial letters K–Z are found in Tab. A3



| Country | EB 87.1 2017 | EB 91.3 2019 | PEW Global Attitudes 2015 | ISSP 2010 | LITSII 2010 | ESS 2016 | Median |
|---|---|---|---|---|---|---|---|
| Malaysia | - | - | 0.76 | - | - | - | 0.76 |
| Malta | 0.77 | 0.86 | - | - | - | - | 0.82 |
| Mexico | - | - | 0.86 | 0.79 | - | - | 0.83 |
| Moldova | - | - | - | - | 0.74 | - | 0.74 |
| Montenegro | - | - | - | - | 0.47 | - | 0.47 |
| New Zealand | - | - | - | 0.60 | - | - | 0.60 |
| Nigeria | - | - | 0.85 | - | - | - | 0.85 |
| Norway | - | - | - | 0.57 | - | 0.50 | 0.53 |
| Pakistan | - | - | 0.71 | - | - | - | 0.71 |
| Palestine | - | - | 0.71 | - | - | - | 0.71 |
| Peru | - | - | 0.90 | - | - | - | 0.90 |
| Philippines | - | - | 0.89 | 0.75 | - | - | 0.82 |
| Poland | 0.68 | 0.74 | 0.64 | - | 0.53 | 0.43 | 0.64 |
| Portugal | 0.78 | 0.81 | - | 0.76 | - | 0.62 | 0.77 |
| Romania | 0.74 | 0.74 | - | - | 0.61 | - | 0.74 |
| Russia | - | - | 0.67 | 0.72 | 0.61 | 0.44 | 0.64 |
| Senegal | - | - | 0.80 | - | - | - | 0.80 |
| Slovenia | 0.75 | 0.77 | - | 0.69 | 0.62 | 0.55 | 0.69 |
| South Africa | - | - | 0.74 | 0.72 | - | - | 0.73 |
| South Korea | - | - | 0.79 | 0.73 | - | - | 0.76 |
| Spain | 0.80 | 0.83 | 0.80 | 0.75 | - | 0.60 | 0.80 |
| Sweden | 0.77 | 0.78 | - | 0.63 | 0.62 | 0.46 | 0.63 |
| Taiwan | - | - | - | 0.78 | - | - | 0.78 |
| Tanzania | - | - | 0.80 | - | - | - | 0.80 |
| Turkey | - | - | 0.73 | 0.81 | 0.55 | - | 0.73 |
| Ukraine | - | - | 0.67 | - | 0.62 | - | 0.64 |
| United Kingdom | 0.67 | 0.76 | 0.70 | 0.62 | 0.52 | 0.48 | 0.65 |
| United States | - | - | 0.66 | 0.61 | - | - | 0.64 |
| Venezuela | - | - | 0.89 | - | - | - | 0.89 |
| Vietnam | - | - | 0.88 | - | - | - | 0.88 |

**Table A3.** Same as Tab. A2 for countries with initial letters K–Z.