# Peer review of "A global threshold model of enabling conditions for social tipping in pro-climate change behaviours – the role of sea-level rise anticipation and climate change concern"

_EGUsphere, 2023_

## Author Response (AR1)

**Comments from Referees and Author Response**

First of all, we thank the reviewers for this constructive and well considered review of our manuscript. We will address these issues raised by each point below. In order to best organise our responses, we have structured these by Reviewer (underlined) and thematic concept being addresses [in brackets]. If similar issues are raised across reviewers, we address these more directly once, and cross-reference of our responses where appropriate for clarity.

**Reviewer #1**

[Why does the paper focus on SLR and not on other impacts? Comment #1]
First, one of the issues raised is that the analysis of this paper focuses on sea-level rise, as SLR is particularly localised. This concern is valid, and well-taken by the authorship team, but this is also true of most other projected future climate impacts. For example, floods, droughts, increased intensity and frequency of hurricanes likely share similar spatial impacts. This comment also raised the issue of the temporal distance between mitigating actions and the manifestation of SLR. This is particularly the motivating reason why we chose to focus this empirical modelling work on SLR, as this is an impact in which there is limited observed data. Rather, the primary question of this research is how a problem like anticipation of future SLR can (or cannot) lead towards increased climate mitigating behaviours and policies. There are many other papers which focus on the impact of extreme weather events (often using observed data after exposure to the event) - but for the future impacts of SLR on climate actions, this is something that can likely only be assessed via a modelling exercise such as this one (particularly on the global scale). Further to this point, we believe that SLR is a particularly unique potential climate impact not only because of the temporal disconnect between mitigating actions now are need to minimise impacts that will manifest in the future, but also importantly, SLR is a very unique potential impact of climate change, as it is particularly persistent. The impact of floods (for example), while potentially causing similar amounts of damage to localised communities and land, are comparatively more temporary. Indeed, we see that in many areas, communities rebuild on the same flood lands within comparatively short periods of time (years). While for SLR, the impacts persist over potentially thousands of years - which for current human societies is effectively permanent. In this case, the impact of SLR is a very risky form of climate change induced event, and one that is comparatively less understood within the social science and empirical literature. Accordingly, we suggest that the revised manuscript would further develop the argument for why SLR is a unique and important climate change induced impact, as well as further developing the existing literature on other impacts to note the differences and similarities with SLR.

[Theory of change and model specification - how do people translate into active population]
We thank the reviewer for bringing up these questions about the 'active' population as well. We agree that concern does not necessarily translate into action, this is similar to the problem of an 'attitudes-behaviour' gap. Our model intends to incorporate this uncertainty, where concern is rather viewed as a necessary - but not sufficient - condition for action. We parameterise the 'potentially active' population using the level of concern for a given country - which simply means that the person has the potential to become activated, but would only do so if a majority of their neighbour nodes become activated. This is clearly reducing a great amount of complexity in human behaviour - but given the model resolution, we explicitly do not intend to deterministically explain how people will become active. Still, we note that such an approach is in line with a comparatively large body of literature on models of social activation that effectively group actors into similar categories like the ones we employ here. Given that our model explores the complexities of the relationship between individual attitudes, network effects, SLR exposure and impact time, we keep the level of specificity for human behaviour as relatively low, as

otherwise, the model components would be very difficult to identify and parameters difficult, if not impossible, to estimate. Thus, human behaviours would likely be at a different depth of specification and out of line with the other model components.

[Why does the paper focus on SLR and not on other impacts? Comment #2]
We thank you for the further comment regarding the choice of SLR as the chosen climate impact we assess in this model. Similar to our response above, we note that SLR is a unique, and comparatively understudied future climate impact. There is a comparatively large literature on the role of extreme weather events affecting attitudes and behaviours - yet even within this literature, the effects vary substantially by operationalization of the independent (event) and dependent (attitude and behaviour). For a recent finding in this regard, Figure 6 and 7s of this pre-print from Quoss and Rudolph (2023) nicely demonstrate substantial noise in the impact of extreme temperature, precipitation and other events shaping attitudes in Switzerland.
That is, we also recognise the difficulty in determining what will actually "cause" social tipping, and in a revised manuscript, would further note the limitations of any extreme event in initiating tipping dynamics - as there is limited empirical evidence that this has yet to occur.
At the same time, we further believe that SLR is an interesting case study, with the potential to motivate action - even though the impacts are indeed likely (hopefully) quite far into the future. For example, a related paper by the authorship group (under review) uses survey data from the United States (a locale with comparatively lower proportion of citizens likely to be affected directly by SLR - Class III), finding that concern about SLR is one of the top factors driving climate change policy support and behavioural willingness (see Smith et al., 2022). Accordingly, this paper focuses on understanding how SLR would link current actions with impacts that will not manifest until generations into the future via the mechanism of anticipation. Indeed, this may be an optimistic - but important question - as actions in response to experienced SLR would already be far too late to substantially mitigate these quasi-permanent changes to vast sections of inhabited land around the world.
Lastly, in this paper we do propose a linkage between SLR and other climate impacts, where anticipation of SLR can provide grounds for event-induced tipping, i.e., the anticipation of SLR can raise concern sufficiently high so that other, more immediate impacts, can kick the system into a new stable state.

[Salience of SLR and how SLR perceptions shape attitudes and behaviors]
This point is very well taken regarding the claim made about the salience of SLR and the reference citation. We have further reviewed the literature on the salience and impact of sea-level rise. Several studies have noted that sea level rise is not particularly well understood by the public (in terms of scientific accuracy, e.g. Thomas et al., 2015, Priestley et. al, 2021), is seen as occurring quite far into the future (Covi et al., 2016), and some studies have also found that SLR does not receive frequent media attention (e.g. Akerloff et al., 2017; 2019). Yet, communicating the risks of SLR can increase climate change attitudes - even amongst more 'hard to reach' audiences such as Republicans in the United States (Bolsen et al., 2018; Smith et al., 2022). Accordingly, we suggest revising the manuscript to further include these relevant literatures, and develop a similar, but re-framed argument:
While SLR may be, at times, be misunderstood and not a priority for some members of the public, it has the potential to be a strong driver of climate change attitudes and behavioral change. Yet, the impact of anticipation of SLR on climate change attitudes and behaviours may be limited if the distance to projected impacts extends far into the future. Accordingly, we model how the likelihood of attitudes and behavioral change relates to differing levels of concern, being spread throughout networked dynamics, at varying levels of impact severity and timeframes.

[Definition of active population]
We agree with this comment and those further raised below - a similar concern to those brought by Reviewer 2 and 4 - regarding the lack of definition for the construct 'active population'. We intended to draw upon an (intentionally simplified) categorization of the population into three groups - certainly active, potentially active, and not active. For the active population in this regard, these would be people that are already engaging to mitigate future climate impacts in their local settings. Given the cross-national design, the 'action' is relative to the needs and capacity for a given setting. This could be political action (e.g. voting behaviour directly for climate change policies, protesting), individual mitigating behaviours (e.g. pro-climate behaviours), or even social actions (spreading of climate communication, shifting norms). Given the resolution and cross-national implementation of this model, we cannot define more specifically what 'action' means, as any given action for any given context and person will a diverse set of motivations and barriers (complexity would need to be explored within a different empirical approach). Furthermore, we also note that the capacity to action for a given individual is likely inverse to the expected impacts of SLR - a distributional inequity posed by climate change. We would propose further revising the manuscript to better elaborate on the definition of climate action, and these distributional impacts.

Furthermore, we also suggest changing the terminology to be more reflective of the social science literature. The most broad phrase that is commonly used is 'pro-climate change behaviors'. Accordingly, would suggest changing the terminology from climate action to 'pro-climate change behaviors' to be more reflective of these literatures.

Also, given the concerns regarding the terminology used for the modelling approach raised by Reviewer 2 [Model terminology and relation to other theories of environmental social change], we suggest revising the title to be "A global threshold model of enabling conditions for social tipping in pro-climate change behaviours – the role of sea-level rise anticipation and climate change concern"

We would further suggest that future research should be engaged to elaborate the mechanisms by which action can develop. Such work would need to be well-tailored to the individual action and setting. There is much work in the regards (largely enacted by environmental social psychologists, public policy researchers and behavioural economists) focusing on specific settings and actions - but disproportionately focusing on Western, wealthier states. Our goal of this modelling exercise was to intentionally to be rather abstract - allowing for cross-national comparisons. But these findings should be seen as in compliment to this rich literature exploring the more specific and diverse mechanisms of environmental change (individually, socially and politically).

Lastly, in regard to the EIB study regarding the desire of 63% of Germans to prefer stricter climate actions, we would rather consider this to be akin to the potentially active population, those willing to act, rather than to the actually active population. Responding positively to a survey question more likely displays a willingness to change, while the actual observed action would be whether this translates into political action (voting). And, as this comment notes, we did not observe tipping like behaviour within the German system (at least not yet). For example, there was a historic increase in support for the Green Party in the 2021 German Federal Election - where climate change was one of the largest issues across all parties. Yet, in the end, the Green Party only received ~15% of the popular vote and seats in Parliament. This resulted in the Green Party being part of the governing coalition (for the first time in 15 years), and likely contributed to many federal actions to combat climate change, but did indeed fall short of rapid, systemic changes. We would read this as evidence of a system that is getting closer towards change, but one that has, as of yet, not tipped.

[Specification of model and policy relevance of findings - Comment #1]

As noted in the previous comment, we also agree with the importance of developing specific mechanisms for how tipping can be induced, and proscribing actions for policy-makers and stakeholders to incentivise these changes. This is important future work - empirical research that would rely upon observed individual data (potentially within experimental settings) to understand how such changes can occur for a given context and towards a specific change. For example, there is another pre-print that focuses on expert elicitations of tipping dynamics towards decarbonization in the UK political system (Smith 2023). Yet, we would also note that defining such a specific mechanism is not the purpose or design of our research question and modelling exercise. This would require a far greater level of specificity, using different methods, and would almost certainly not be generalizable across such a cross-national design. We would revise our future research statement in a revised manuscript to make this call for further research more clear and direct.

Yet, in terms of political relevance from our modelling approach, we emphasise the role of creating enabling conditions for change to occur (e.g. Tabara et al., 2018; 2023). We find, that on it's own, concern may be insufficient to induce tipping dynamics, but rather, provides the necessary conditions for such to change to take place (either induced by other events, policy changes, action etc). That it, concern pushes the system closer towards the threshold, such that tipping can occur.

[Specification of model and policy relevance of findings - Comment #2]
We further thank you for the comments regarding the specificity and complexity of this model. As we have noted above, we very much agree with the need for further research (across many disciplines) that further explores this complexity. For example, within a modelling exercise, research on interactions using a similar modelling design could also be engaged. Members of the authorship team have also engaged in such work using a similar model (e.g. Müller et al., 2021). We will further elaborate on different types of future research that could expand upon these findings (from an experimental, observational and modelling approach) within the revised manuscript.

We would also contend that, like most models, some of the conclusions are more directly related to the modelling design (that is, the more people are exposed to direct impacts of SLR the more likely they are to act). Yet, we contend that others are not as "obvious". For example, the differing role of concern in shaping the likelihood of action. This is illuminated within the three classes of tipping section of the results - where we find that yes, within class I, the results are rather expected, but in classes II and III, there are interesting interactions between SLR and concern that are exposed, which we believe are non-trivial and not necessarily anticipated. We would suggest revising the manuscript to further emphasise the results that go beyond the "top line" findings, that may be more related to the modelling design itself.

**Reviewer #2**
We thank the reviewer for the overall positive review of our paper. We have responded to each of these comments and suggestions below, organised by topic.

[Why focus on 5 exemplary countries in Figure 1?]
The reason for focussing on the five countries with the largest shares of GHG emission in the first part of the manuscript's results section is twofold: First, we acknowledge that those countries are potentially also among the ones where effective climate policy might have the most substantial impact on global carbon emissions. At the same time, it is reasonable to assume that these countries do in fact possess the political capacities to lead others towards joining into collaborative efforts to mitigate climate change. Second, quantifying which countries are most at risk with respect to human population or socio-economic impacts is a highly multidimensional problem that is still subject of current research. Had we attempted to put our focus according to

any of these dimensions our choice would have likely become very subjective. Ultimately, we note that previous works on the interrelation between future climate variability and policy have opted for similarly focusing on the five top emitters (Ricke & Caldeira, 2014). For the revision we propose to add a statement that justifies our choice of presented countries using the arguments above to the beginning of Sec. 4.1 in the manuscript.

[Spelling and technical issues]
We agree with the reviewer that the notion of a qualitative state in social systems deserves further clarification.We therefore propose to add examples from technology adoption, political systems, and environmental behaviour to the respective section in the introduction of the paper. We thank the reviewer for highlighting a few typographic errors, which we will fix during the revision.
We will change the sentence on line 48 to 'The ultimate trigger of social tipping processes can come from forces originating from either natural or social systems [...]'.

[Individual and societal level changes]
Yes, we agree that individuals are part of social systems. For this analysis, we focus on a theory of change that originates (is triggered) from individual (micro-) level dynamics, as opposed to from systemic level changes (e.g. policy change). That is, we explore how these individual changes can enact system regime changes, which themselves reshape the individual actions nested within these regimes. This approach is situated directly within the "social tipping" theory of change elaborated within the paper, but should naturally be interpreted as complimentary, and part of, broader mechanisms towards how such rapidly shifts could occur. The interplay between micro-macro-micro changes are a foundation of social analyses looking at the co-evolution of structure and agency (e.g. "Coleman's Boat", Coleman 1994). In a revised version of the manuscript, we would make the connection between micro actions and resulting macro level changes more clear and linking to relevant literature on such theories of change.

[Uncertainty of changes resulting from climate impacts]
This is a very good point that we agree with. As we note in our response to Reviewer #1 above, there is much hetereogeniety in the effect of climate impacts on climate change attitudes and behaviour, and as this comment correctly notes, not all of these changes may be normatively 'positive'. Furthermore, there is recent literature exploring the risks of societal tipping dynamics (e.g. Milkoreit 2023), which emphasizes the inherent danger of rapid social change, which is a very chaotic event and the outcomes are far from determined. Indeed, contemporary examples used to identify social tipping (e.g. Arab Spring) have not always resulted in long-term, "positive", change within these societies. We suggest revising the discussion and theoretical background to emphasize the uncertainty not only in the effects of extreme events, but the direction of the resulting changes.

[Model terminology and relation to other theories of environmental social change]
This is a good point that is also raised in regard to the relation of the theory of change in this model to other forms of change that have been applied to environmental behaviours. The current version of the manuscript does not clearly elaborate the connections between social tipping and established theories of change - as it is more focused on applying these concepts in the modelling exercise as opposed to theoretically explaining them. But here, we would highlight the direct connections between classical threshold models of behavioural change (e.g. Granovetter 1978, Schelling 1972) alongside those focused on developing a micro-based foundation for social change (as noted above, and in line with the VBN theory noted here). This is not as directly connected to TBP based mechanisms, or other theoretical explanations of the attitude-behaviours gap (e.g. Kollmuss and Agyemann, 2002; Diekmann and Preisendörfer,

2003), as these are rather understood by the authorship team as potential barriers and limitations to the theory of change explored by this model. We suggest revising the manuscript to make the connections between social tipping literature and other proposed theories of environmental change (individual and structural) more clear.

We also suggest changing the name of the model to be "network-based threshold model for social tipping" to further clarify these connections to previous literature (Wiedermann et al., 2020) and reduce confusion over this modelling approach.

Lastly, as noted in our response to Reviewer 1 [Definition of active population], we would also suggest changing the title to reflect the new terminology of the model, as well as the proposed change in terminology for climate action: "A global threshold model of enabling conditions for social tipping in pro-climate change behaviours – the role of sea-level rise anticipation and climate change concern"

[Connections to resource mobilization theory]

We agree with the reviewer that resource mobilization theory has already been around for half a century. In our original paper (Wiedermann et al., 2020) we motivate our choice of the model by acknowledging that spreading of opinions and behaviours can often be explained through complex contagion (Watts, 2002; Centola et al., 2015 & 2018). At the same time, recent studies have investigated the role of 'instigators' (Singh et al., 2013) or 'immune' individuals (Karsai et al., 2016) in addition to the contingent population, as such groups are commonly discussed in the literature and seem apparent in real-world situations. Notably, such distinctions, even though they were only studied quantitatively in the recent decade, align very well with the attribution of groups in resource mobilisation theory (RMT). In this sense, we only draw the terminology from RMT without inferring any causal mechanism related to that theory. We think that using a tripartite categorisation is appropriate as it approximates relevant actors in any social movement in a meaningful way without the need for specifying more groups than necessary: those that will act, those that are willing to act and those that will never act.

We propose to rework our manuscript so that it becomes clear that all we draw from RMT is the terminology of groups while the rest follows from literature that is much more recent.

[Connection between climate change concern and SLR perceptions]

We completely agree with the reviewer that concern can not necessarily be linked to awareness or proper risk assessment. However, it seems that there is a misunderstanding at play that we will attempt to clarify in the revision of the manuscript.

The core idea of differentiating between potentially active and certainly active individuals is that the former don't necessarily need to be directly affected by sea-level rise. Rather, it is the certainly active population, i.e., those that are or will be affected, that we assume to trigger a movement so that the potentially active individuals will join in, regardless their specific locale in a country. In other words, in order to instigate a movement, an individual needs to be both affected and concerned. Only then will those that are concerned, but not affected, join into a certain climate action based on social contagion and peer pressure.

We also agree that concern is not even distributed within countries. Indeed, there may even be more intra-country variance than across countries. But, there is the potential for spillovers within a country, for example, a person living in Berlin could be very concerned about SLR (but not directly affected). Yet, the person is connected to an individual in Hamburg who is concerned and affected, which could trigger both towards action. We suggest including this as a limitation and suggestion for future research in the revised manuscript.

Further, this comment notes that we use climate change concern, and not SLR risk perceptions, to operationalise the potentially active population. This is rather a product of the data limitations, where questions about the perceived risks associated with climate change are often asked more broadly, and less frequently in respect to specific impacts (such as SLR). While we

agree that we do not directly capture SLR perceptions in the data used for this study, climate change concern can function as an appropriate "best available data" proxy. For example, recent research conducted by the authorship team suggests a close relationship between anticipation of SLR and climate change concern, where we find a strong, positive correlation r=0.76 in a recent sample of US adults (see Figure A1 in Smith et al., 2022). Accordingly, we would add this information as part of the revised limitations mentioned above.

**Reviewer #3**
Thank you as well for these constructive comments, which we respond to by each topic raised below.

[SLR as chosen impact and distributional issues]
Thank you very much for the supportive comments. We very much agree with the potential to expand this analysis to other climate impacts. This is something we will further emphasise in the revised manuscript. In our response to the other expert reviews, we also note how we will further elaborate the rationale for why we focus on SLR.
We also agree with the need to focus on inequalities associated with these impacts. While the model does not allow for the specificity to talk about specific societal subgroups within a given country (e.g. race, class) - we can make a clear link between the likelihood of people to be exposed to SLR (and related impacts) and the comparative diminished capacity/agency of these people to respond. This would speak directly to the concern raised, that the people most likely to 'tip' are less likely to have the capacity to act. This is something that we view as important to highlight, and will expand upon in the revised discussion.

[Terminology and Specific Technical Changes]
We thank you for these more specific comments as well. We will address each of these accordingly within the revised manuscript.

**Reviewer 4:**
Lastly, we greatly appreciate the feedback provided as well. Similar to the other comments, we have responded by topic below, as well with linkages to other responses to similar comments we have addressed above.

[SLR as Chosen Climate Impact]
Thank you very much for this comment. This is something that is shared commonly across many of the reviewers. For the sake of parsimony, we will not repeat our responses here, but would rather refer to our replies to Reviewer #1 [Why does the paper focus on SLR and not on other impacts? Comment #1] and [Why does the paper focus on SLR and not on other impacts? Comment #2] and to our response to Reviewer #2 [Uncertainty of changes resulting from climate impacts].

[Heterogeneity in the effect of concern]
Thank you for raising this point, we very much agree with this. We also note some of these issues about the variance in the effect of concern in shaping climate actions in our response to Reviewer #1 [Salience of SLR and how SLR perceptions shape attitudes and behaviors] and Reviewer #2 [Connection between climate change concern and SLR perceptions] (amongst other comments above). We very much appreciate the further suggestion to expand the discussion on what this means in terms of political relevance. If a system has higher levels of concern, we find that it has the necessary conditions for change, and in this case, other efforts (such as pressuring politicians) could be successful to push towards a tipped state. But of course, what action can be

taken to "trigger" the social tipping depends on the local context. We suggest expanding the discussion in the revised manuscript to make these recommendations and relevance more clear.

[Defining Action]
Thank you for this comment as well, this was similarly shared by Reviewer #1 [Definition of active population]. We will refine this definition in the revised manuscript in line with our response to Reviewer #1, but in a particular response to this comment, we would consider adaptation to SLR to likely not be included as action, as in this context, this likely means that the impact of SLR has already occurred, and is likely unchangeable during the normal lifetime of humans. Rather, we are interested in how people can be motivated to act to minimise the risks of being exposed to SLR - so focusing on mitigating actions.

[Heterogeneity across subgroups within population]
We also agree with some of the concerns raised here about how the mechanism of never active->potentially active->certainly active may vary for certain subgroups within a population. Given the model resolution, such intra-country dynamics are not included in these analyses. Certainly, these could work both ways - where some countries will have sub-groups that are more likely to act and some that are not. We agree that this is a limitation that would be discussed in a future revision of the paper, as well as a call for further research into within country dynamics, which could build upon this assessment.

[Exclusion of lower elevation regions]
We had excluded lower lying regions (e.g. Azerbijian, Kazakhstan and the Netherlands) as well as those that are situated inland (e.g. Mongolia, Austria) as these present more 'extreme' case that are likely far outside of data distribution of the remaining sample. Particularly in the case of lower-lying countries, we consider these to be more specific cases, that are not likely as well explained by the model. This is an approach that has also been adopted by similar modelling exercises (e.g. Marzeion and Levermann, 2014). We would suggest revising the manuscript to make this justification more clear, and if the reviewer believes appropriate, present additional analyses including the data from lower lying regions as a robustness check.

[Connection to theories of change]
Thank you for the final comment on the discussion of sensitive interventions and other theories of change. This is similar to a comment raised by Reviewer #1 [Model terminology and relation to other theories of environmental social change]. Accordingly, we suggest revising this section in the discussion, to be more clear how social tipping relates to other theories of change, and how the proposed change would occur (bottom-up). We believe this would make these statements more clear to the reader.

**Changes in manuscript**

Based upon the reviewer comments, we have made a number of changes and edits to the revised manuscript. These are listed below, in terms of order in which the changes appear:

1. Changed the title to "A global threshold model of enabling conditions for social tipping in pro-climate change behaviours – the role of sea-level rise anticipation and climate change concern"
2. Revised abstract for language and clarity
3. Changed terminology for the modelling approach "network-based threshold model for social tipping"
4. Revised introduction to accentuate importance of SLR
5. Changed terminology for climate actions to "pro-climate change behaviours"
6. Revised subsection on social tipping, further accentuating how social tipping relates to other theories of change, and distinguishing the importance of enabling conditions.
7. Revised the subsection on climate change concern and how higher levels of concern can provide a necesarry condition for pro-climate change behaviour adoption
8. Revised section on anticipation of SLR to reflect extant literature on perceptions and knowledge of SLR, as well as make the case for why it is importance to model how anticipation of SLR can shape behaviours
9. Revised modelling description for clarity, to make sure terms are properly introduced and distinctly described in the methods section
10. Added in justification for why we focus on 5 exemplary countries in Figure 5
11. Expanded limitations section to include:
    a. Relation of behaviours in response to SLR to extant literature on behaviours in response to extreme weather events
    b. Reflection on the role of climate change concern shares in the model, and the relation between climate change concern and anticipation of SLR
    c. Explanation and limitations of excluding low lying and inland countries from model
12. Expanded outlook section to include:
    a. Call for further research that identifies the inequalities in populations that are most likely to be exposed to SLR and their capacity to make changes
    b. Call for further research to develop more specific mechanisms for social tipping processes.
    c. Reflection on recent criticisms of social tipping processes, and how this manuscript should be interpreted in light of concerns

**References:**

Akerlof, K., Covi, M., & Rohring, E. (2017). Communicating sea level rise. In Oxford Research Encyclopedia of Climate Science.

Akerlof, K., Merrill, J., Yusuf, J. E., Covi, M., & Rohring, E. (2019). Key beliefs and attitudes for sea-level rise policy. Coastal Management, 47(4), 406-428.

Bolsen, T., Kingsland, J., & Palm, R. (2018). The impact of frames highlighting coastal flooding in the USA on climate change beliefs. Climatic Change, 147, 359-368.

Coleman, J. S. (1994). Foundations of social theory. Harvard university press.

Covi, M. P., & Kain, D. J. (2016). Sea-level rise risk communication: Public understanding, risk perception, and attitudes about information. Environmental Communication, 10(5), 612-633.

Diekmann, A., & Preisendörfer, P. (2003). Green and greenback: The behavioral effects of environmental attitudes in low-cost and high-cost situations. Rationality and Society, 15(4), 441-472.

Kollmuss, A., & Agyeman, J. (2002). Mind the gap: why do people act environmentally and what are the barriers to pro-environmental behavior?. Environmental education research, 8(3), 239-260.

Marzeion, B., & Levermann, A. (2014). Loss of cultural world heritage and currently inhabited places to sea-level rise. Environmental Research Letters, 9(3), 034001.

Milkoreit, M. (2023). Social tipping points everywhere?—Patterns and risks of overuse. Wiley Interdisciplinary Reviews: Climate Change, 14(2), e813.

Müller, P. M., Heitzig, J., Kurths, J., Lüdge, K., & Wiedermann, M. (2021). Anticipation-induced social tipping: can the environment be stabilised by social dynamics?. The European Physical Journal Special Topics, 230(16-17), 3189-3199.

Priestley, R. K., Heine, Z., & Milfont, T. L. (2021). Public understanding of climate change-related sea-level rise. PLoS One, 16(7), e0254348.

Quoß, F., & Rudolph, L. (2022). Operationalisation matters: Weather extremes as noisy natural experiment show no influence on political attitudes. https://osf.io/preprints/osf/m5rz8

Smith, E. K., Eder, C., Donges, J. F., Heitzig, J., Katsanidou, A., Wiedermann, M., & Winkelmann, R. (2022). Domino Effects in the Earth System-The role of wanted social tipping points. OSF Preprints.

Smith, S. R. (2023). Enabling a political tipping point for rapid decarbonisation in the United Kingdom. EGUsphere, 2023, 1-21.

Tàbara, J. D., Frantzeskaki, N., Hölscher, K., Pedde, S., Kok, K., Lamperti, F., ... & Berry, P. (2018). Positive tipping points in a rapidly warming world. Current Opinion in Environmental Sustainability, 31, 120-129.

Tàbara, J. D. (2023). Regenerative sustainability. A relational model of possibilities for the emergence of positive tipping points. Environmental Sociology, 9(4), 366-385.

Thomas, M., Pidgeon, N., Whitmarsh, L., & Ballinger, R. (2015). Mental models of sea-level change: A mixed methods analysis on the Severn Estuary, UK. Global Environmental Change, 33, 71-82.

Wiedermann, M., Smith, E. K., Heitzig, J., & Donges, J. F. (2020). A network-based microfoundation of Granovetter's threshold model for social tipping. Scientific reports, 10(1), 11202.